# OpenReview forum: "A Hitchhiker's Guide to Scaling Law Estimation"
_ICML.cc/2025/Conference — ICML 2025 poster_

### Official Review · Reviewer_CJfP · 2025-03-09

**Overall Recommendation:** 4

**Summary:**

This work investigates the challenge of fitting scaling laws in the language model domain, in particular:
- How accurate we should expect scaling laws to be? (The authors show ~ 4% Absolute Relative Error (ARE) at best).
- How does the shape of scaling laws vary with architecture, in order for the community to efficiently use scaling laws to propose architectures modifications? (The authors show "similar enough to be useful and transferrable").
- Should we only use final checkpoints? (The authors show we should use checkpoints from ~> 30% of training data, and that only using final checkpoints can be detrimental).
- How large should models be? (The authors show a nuanced trade-off between variance of larger models and proximity of larger models to extrapolated models of interest).

The authors do the above through a meta-analysis of existing open pretrained Language Models (LMs).

### Update after rebuttal

I increased my score to 4: accept. The critical issues I had with the paper (the numerical values of the coefficients) were addressed, and turned out to be only a presentation issue. I see no issues in accepting this work, as it is correct to the best of my understanding, and can provide a valuable resource for the community.

**Claims And Evidence:**

### Claim 1: Scaling laws predict at best 4% ARE

#### Supported (previously partially supported)

Edit: authors clarified difference between residuals and ARE.

Section 4 of the current paper provides a good discussion of the source for this value. This Figure 4, where the typical residual has an absolute value of 0.02. With Chinchilla losses ~2, this corresponds to an ARE of ~ 1%. This analysis is open source and reproducible. The authors should comment on the discrepancy between their result and the result in [1].

### Claim 2:  Scaling laws can be strategically reused across model families

#### Supported (was previously supported)

Edit: authors clarified misunderstandings here, including reporting log values.

In Section 5, the authors note that $E, A, B, \alpha, \beta$  vary significantly over model families (Figure 3). I find the numerical values in Figure 3 concerning:
1. $A$ and $B$ are more than an order of magnitude different to typically reported results (see e.g. [1]) for very similar setups.
2. Negative values of $E$ do not correspond to a well-behaved asymptotic limit, as cross-entropy loss is lower bounded from zero.
This hints at an issue related to the optimization process for the parameters. I note that the scaling law estimation is done using `curve_fit` in `sklearn` (section 3.2), and the authors comment they could not get the L-BFGS based solver to be stable. Additionally, the authors optimize square loss, instead of Huber loss. They state that Huber loss produces similar trends as stable loss (Appendix E), yet, they also see throughout their work the effect of single poor runs dominating scaling law fitting (Section 7). I suggest the authors investigate the code provided in [1] as a way of robustly identifying scaling law coefficients, which can be sensitive to outliers and initial conditions, and may prevent robust identification. Because of potential numerical challenges, this claim, and remaining claims depending on derived scaling laws are only partially supported.

Beyond the above, the authors take pre-existing $E, A, \alpha$ values, and fit only the data term $B, \beta$ values. In Figure 6 the corresponding error is shown.

It is unclear what fitting process is being performed here to produce the shown errors. Which cell(s) correspond to the fitting of a single model? Does it matter which single model? It would be useful for the authors to explicitly describe the fitting process for this Figure in the accompanying Appendix.

What are the resulting $B,\beta$ values? Can the authors also provide the Absolute Relative Error (ARE) for the model at the end of training, rather than an average over the $\geq 30\%$ checkpoints, as the latter value is the primary one of interest for drawing conclusions about.

### Claim 3: Use $\geq 30\%$ checkpoints, not only final checkpoints

#### Supported (was partially supported)

Edit: authors clarified number of runs versus number of checkpoints understanding. Some comments about Chinchilla are still outstanding, however, the required data for that analysis is not available, and not a blocker for the main conclusions/contributions of the paper.

In Section 6.1 and Figure 4, the authors show how varying the checkpoints used for the fit changes the ARE. The behavior shown in Figure 4 does support the authors claims.

The baseline number of models being used for the fit in each plot in Figure 4 is low. Can the authors explain how it is possible to achieve the fitting of an equation with five free parameters $E, A, B, \alpha, \beta$ to ~3 models (Figure 4b, top left cell) such that the resulting scaling law reliably generalizes? Is this true for all choice of 3 models? What is the variance of the scaling result over resampling of the chosen 3 models? It would be useful the the authors to perform the relevant bootstrap analysis here.

Does the claim remain true when a significant number of models are included in the fit. E.g. Chinchilla uses >200 models in the fit. Is it still beneficial to include intermediate checkpoints in this scenario, or do the checkpoints now result in a biased estimate?

In addition to the above question regarding robustness of generalization, partially supported because of numerical issues discussed in Claim 2.

### Claim 4: There is a trade-off between using fewer larger models or more smaller models

#### Supported (was partially supported)

Edit: issues above now resolved.

Section 7 and Figure 2 support the authors claims. Partially supported because of numerical issues discussed in Claim 2.

[1] Chinchilla Scaling: A replication attempt, https://arxiv.org/abs/2404.10102

**Essential References Not Discussed:**

The re-usability of scaling laws across model families to iterate on architecture choices is well-captured by [3], which builds the scaling law into a hyperparameter search procedure, and is relevant for the ideas relating to Claim 2.

[3] https://arxiv.org/abs/2302.00441 Scaling Laws for Hyperparameter Optimization

**Experimental Designs Or Analyses:**

I checked the experimental design in the paper. The meta-analysis experimental design is sensible.

There are a number of ways the experimental design could have been improved:
- It would be beneficial if the authors more fully detailed their precise optimization scenario (learning rates used)
- The data from [1] should have also been included, as it provides a large number of final checkpoints within a single family, an important use case not shared with other data sources used by the investigation.
- The numerical methods chosen (see discussion in Claim 2 above) are potentially problematic, and yield coefficients that don't correspond to well-behaved scaling laws (Fig 3c) or coefficients consistent with prior approaches. The issues faced by the L-BFGS solver should be understood. (For example, did the authors provide the Jacobian to the solver?).
- There is no discussion of which combinations of model size $N$ and data $D$ to pick to fit a scaling law. Some choices of $N$ and $D$ yield pathological solutions and do not allow Identifiability. One common problem is the temptation to use only Chinchilla-optimal models for fitting the scaling laws. In which case, $N = k D$ and the scaling law can be written as a one-dimensional scaling law in either $N$ or $D$, which produces redundancies around the solutions for $A,\alpha$ and $B,\beta$ as they cannot be disentangled. Doing a decomposition of the type described in Section 9 would reveal that 3 components in this case would explain most of the variance, as the authors find. IsoFLOP protocols, as in Chinchilla, are designed to mitigate this pathology/redundancy. Two questions are then:
	- If we use IsoFLOP models to fit the scaling law, are the conclusions of Section 9 still true?
	- Is the observation that using checkpoints beyond the final checkpoint is beneficial an artifact of the models being analyzed satisfying $N\propto D$, and the utility from early checkpoints arising due to requiring a disentangling between $A,\alpha$ and $B,\beta$, which might then disappear in the case where the models being analyzed are IsoFLOP?
- More generally, the paper and experimental design would benefit from a discussion regarding which $(N,D)$ combinations to choose subject to a compute budget, taking into account this identifiability issue.

[1] Chinchilla Scaling: A replication attempt, https://arxiv.org/abs/2404.10102

**Methods And Evaluation Criteria:**

The approach of a meta-analysis of many scaling studies is a sensible approach. The evaluation criteria is reasonable, and corresponds to questions of interest in the scaling laws community. The baseline scaling laws chosen in Section 5.2 are reasonable.

**Other Comments Or Suggestions:**

Figure 6. Explain what "Percentage" means in caption, to help reader (prevent requirement from re-referencing main text.)

**Other Strengths And Weaknesses:**

The paper is a joy to read. It is clearly written, and provides a clean set of definitions and notations that, on their own will aid in community discussion of scaling laws.

The open nature of the project further increases its utility, both in the provided code, and the assembled data from many sources.

**Questions For Authors:**

Edit: Majority of questions responded to. Some outstanding questions corresponding to "model training strategies" for compute-optimal identification of scaling laws, but not a blocker for acceptance.

1. Can you explain the discrepancy between the 4% ARE discussed in Section 4, and the 1-2% ARE observed in Chinchilla [1]? [this discrepancy should be understood as it conflicts with one of the primary contributions in the scaling law field]
2. Optimization of scaling law related [the potential optimization difficulties that may be present in the paper raise questions about the primary claims in the paper, as those claims follow from the optimization, i.e. scaling law fitting]:
	1. Can you explain the negative values for $E$ in Figure 3?
	2. Can you explain the discrepancy of > an order of magnitude between your values for $A$ and $B$ and those of prior work, e.g. [1]?
	3. Do you still see the effect of poor runs affecting scaling law fits (Section 7) in your Huber Loss analysis?
	4. Why didn't you use Huber loss throughout (which would be standard standard scaling law practice), instead of square loss? Do you have a quantification of the difference?
	5. How stable is your scaling law fitting procedure under i) resampling experimental data for fitting, and ii) change of initial conditions for the fit? (i.e. what are your confidence intervals under bootstrap?)
	6. In your failed L-BFGS solver, did you provide the Jacobian? If not, this can be done using autograd, (e.g. JAX), and can greatly help the solver.
3. What happens to Claim 3 if you increase the number of models available (e.g. in [1], >200 models were used)? If the conclusion/guidance the same, or now in favor of using only final checkpoints? If the latter, at what point can we discard earlier checkpoints?
4. Do the conclusions of Section 9 still hold if the scaling laws are fit to an IsoFLOP experimental protocol? (this can be tested synthetically)
5. More broadly, the paper mostly works with models of different sizes that happen to be trained on certain amounts of data. Can the authors comment on the importance of the interaction between $N$ and $D$, and how certain combinations of $N$ and $D$ for a model may be optimal given a compute budget, and if any of their conclusions would change if a strategic choice of models with combinations of $N$ and $D$ are trained by a practitionr?

[1] Chinchilla Scaling: A replication attempt, https://arxiv.org/abs/2404.10102

**Relation To Broader Scientific Literature:**

The work contains many contributions that, assuming correct, are extremely valuable for the scaling community. In particular, Claim 2 "Scaling laws can be strategically reused across model families", would enable more rapid investigation of scalable methods. It would be great to see a deeper dive into this particular claim.

Claim 3 is consistent with findings in [2] as is discussed in the main text.

Claim 1 has potential conflicts with existing literature (see above). Claim 4 - that it may be more effective to train a number of small models than a single large model - is interesting, but to be made fully usable, the authors should provide some guidance/rules-of-thumb for the reader if possible.

[2] https://arxiv.org/abs/2406.19146v1 Resolving Discrepancies in Compute-Optimal Scaling of Language Models

**Theoretical Claims:**

The paper contains no theoretical claims.

---

> ### Author Rebuttal · Authors · 2025-04-01
>
> We thank the reviewer for their interest and for deeming our work “extremely valuable” as well as for the sincere effort in ensuring all the details are in place. We believe the two most pressing issues found are, in fact, simpler to explain than one might expect and hope the reviewer will agree with us on this point.
>
>
> Regarding the ARE vs. residuals: the main difference is that residuals explain the ability to fit the training data (goodness of fit), and in the ARE we measure the prediction error on a test model. We will clarify this in the paper to avoid confusion with related works (such as the one mentioned).
>
>
> Regarding estimated parameter values: as noted in our response to Hw8m  noted, this was a typo: we estimate (and plot) the log of these scaling parameters (as in previous work) to improve stability and enforce positivity. .  Thus, this is a minor error in presentation not an issue with the experimentation. We will fix it in the final version of the paper.  We will also explain more about the rest of the fitting details as requested. For your interest in fitting pre-existing parameters, we do fit the token parameters, and regarding comparing only to the last model, results are very similar, just slightly more noisy when compared per cell in the tables.
>
>
> Regarding the number of checkpoints, note that the number of checkpoints used is much larger than 3: it is the number of unique pretraining *runs* that is relatively small, we discuss the contribution of single seeds and the effect or number of model sizes used in Section 7,8 .
> We agree that the question about chinchilla is interesting, but as they did not release the necessary checkpoints to answer it, we can only assume based on the rest of the experiments that it would not change much (as they have a lot of data already). We did, however, run experiments with the data they did offer (or that we can extract as done in the mentioned paper). The results match the rest of our findings in the paper (when relevant).
> Thanks for the reference to the hyperparameter search, we missed it and will discuss it in the final version.
>
>
> Regarding the questions about fitting with L-BFGS: it is not that one cannot use L-BFGS; with appropriate hyperparameter choices, it does indeed converge. But, the optimization scheme used in the paper was equally effective and required less tuning. We did have various initial experiments with L-BFGS, and they ended up with similar results. We will clarify this in the paper.
>
>
> Regarding the question about the interplay between choices for data, model size and the scaling laws. We believe that the impact here is not great, because we also fit scaling laws on the partial training runs. This means that models were often far from optimality. The only attribute they still share is that the scheduler might converge at the optimal point. Note that not all papers we used even rely on compute optimal thresholds (For example, Pythia is under-trained, the overtrain paper trains models on varying amounts of data, and OLMO models trained with the same amount of tokens regardless of the model size)

---

> > ### Comment · Reviewer_CJfP · 2025-04-03
> >
> > Thank you for responding to myself and the other reviewers. My review and score have been updated correspondingly.

---

### Official Review · Reviewer_buHt · 2025-03-11

**Overall Recommendation:** 3

**Summary:**

The authors address challenges in scaling law estimation by compiling a dataset of training losses and evaluations from 485 pretrained language models. Through extensive empirical analyses, they establish concrete best-practice guidelines for efficiently predicting the performance of new, larger models without fully training them. Key recommendations include leveraging intermediate training checkpoints, excluding unstable early-stage training data, using multiple smaller models to reduce estimation variability, and generalizing scaling parameters across related model families. Overall, the paper serves as a practical guide for efficiently and reliably estimating scaling laws in large language model training.

**Claims And Evidence:**

The empirical evaluation supporting the paper’s main claims is solid, demonstrating that their practical guidelines—such as using intermediate checkpoints, excluding early-stage noisy data, and training multiple smaller models—improve prediction accuracy within the studied scenarios.

However, the paper's findings depend heavily on implicit assumptions about training dynamics, particularly that hyperparameter choices (e.g., learning rate schedules) do not significantly disrupt scaling behavior. Since the authors do not systematically examine how these hyperparameters affect their conclusions, the generality and practicality of these guidelines beyond the presented setup remain uncertain. Additionally, without deeper theoretical insights or a clear understanding of what fundamentally constitutes robust scaling behavior, the long-term applicability of these empirical guidelines may be limited.

**Essential References Not Discussed:**

The related work is well-grounded and sufficiently supports the paper’s contributions.

**Experimental Designs Or Analyses:**

The experimental designs and analyses are generally sound, systematically varying key factors such as model size, dataset coverage, and checkpoints. However, the authors do not fully explore sensitivity to variations such as random seeds or specific hyperparameter choices, which could significantly affect results. While their recommendation to use multiple smaller models appears beneficial in their tested scenarios, further validation of robustness to these factors would strengthen confidence in the general applicability of their conclusions.

**Methods And Evaluation Criteria:**

The proposed methods and evaluation criteria are appropriate and well-chosen for estimating scaling laws. The authors cover nearly all major publicly available LLMs. Their evaluation criterion is sensible too.

**Other Comments Or Suggestions:**

-

**Other Strengths And Weaknesses:**

Strengths:

1. Conducts comprehensive empirical analysis addressing practical questions in scaling law estimation.
2. Offers clear, actionable guidelines (e.g., using intermediate checkpoints, excluding noisy early training).
3. Clearly structured and accessible, effectively using visualizations and succinct takeaways.

Weaknesses:

1. Contributions are primarily empirical with limited theoretical novelty.
2. Focuses exclusively on current families of language models, leaving general applicability uncertain.
3. Methodological assumption that a single scaling law fits entire training trajectories overlooks hyperparameter influence.

**Questions For Authors:**

1. Have you investigated how different learning rate schedules affect scaling-law estimation accuracy, particularly when using intermediate checkpoints?

2. How are the hyperparameters (like the learning rate) selected?

3. Can you provide additional evidence or analyses on the sensitivity of your guidelines to hyperparameter variations and random seeds?

**Relation To Broader Scientific Literature:**

The paper extends prior scaling law research by focusing on practical estimation strategies rather than deriving new theoretical formulations. It consolidates insights across a broad set of models, validating empirical heuristics like using intermediate checkpoints and model family generalization.

**Theoretical Claims:**

The contributions are empirical.

---

> ### Author Rebuttal · Authors · 2025-04-01
>
> We thank the reviewer for the comments and feedback and for proposing that the paper should be accepted. In essence, we also agree that this paper is a meta-analysis trying to figure out what current trends tell us about the world rather than prove that this has to be the case forever. We do believe, however, that with the vast number of cases presented, most of the results will hold until a significant paradigm shift would enter the way we pretrain models, making the knowledge shared here valuable. We believe that the answer to question 2, might shed a lot of light on that issue as well.
>
> We address the weaknesses mentioned:
> 1. “Contributions are primarily empirical with limited theoretical novelty.”
> The contributions are indeed not a mathematical theory, while guarantees would be nice, we see scientific value also in empirical knowledge.
> 1. “Focuses exclusively on current families of language models, leaving general applicability uncertain.”
> We see the replication across settings as a strength of this paper rather than a weakness. While most scaling law papers deal with a single family of models and describe a “law of nature” observed in it, we describe what is consistent across settings and is hence expected to generalize.
> 1. “Methodological assumption that a single scaling law fits entire training trajectories overlooks hyperparameter influence.”
> We kindly disagree that this is an assumption we make. Instead, we hypothesize that this is the case and bring supportive evidence that this is a useful simplification of the setting and that the benefits from the additional data outweigh the downsides, across many models trained in very different hyperparameter settings.
>
> Responses to questions asked:
> 1. “Have you investigated how different learning rate schedules affect scaling-law estimation accuracy, particularly when using intermediate checkpoints? “
> To some extent. Our results aggregate results across multiple models trained with different LR schedules; characterizing their effects in a fine-grained way would be an interesting question for followup work (and one only possible to answer using the dataset released with this paper).
> 1. “How are the hyperparameters (like the learning rate) selected?”
> We do not select any hyperparameters throughout the paper. All the models we report are hyperparameter choices from pretrained models that publicly shared their pretraining loss, like GPT 3 (we extracted) or Pythia. This is also the reason why we expect everything to generalize, as, in essence, it already generalizes across papers and models that each had their own set of choices.
> 1. Can you provide additional evidence or analyses on the sensitivity of your guidelines to hyperparameter variations and random seeds?
> If you consider the fact that every model size is initiated differently, and has the data shuffled separately, and that we have models across different papers. Then, you can see that seeds are changing a lot here and we account for them. See paragraph from line 347 on for the main place where this appears in the context of the paper.

---

### Official Review · Reviewer_43ps · 2025-03-14

**Overall Recommendation:** 3

**Summary:**

This paper provides a comprehensive analysis of scaling laws in large language model (LLM) training, focusing on how to estimate and interpret scaling laws effectively. The authors construct and release a large-scale dataset containing training losses and evaluations from several pre-trained models, enabling them to derive over 1000 scaling laws. The study presents best practices for cost-efficient scaling law estimation and discusses trade-offs between the number of preliminary models, their sizes, and the dataset used.

**Claims And Evidence:**

* The paper makes strong, well-supported claims backed by demonstrating scaling behavior across multiple model families.
* The claim that intermediate checkpoints improve scaling law estimation is well-supported with empirical evidence.
* The assertion that smaller models can sometimes provide better estimates than a single large model is backed by statistical analysis.

**Essential References Not Discussed:**

N/A

**Experimental Designs Or Analyses:**

The statistical robustness of their findings is high, as it aggregates data across 1000+ scaling laws. Extrapolation to unseen models is tested.

Inference efficiency comparisons are not provided.

**Methods And Evaluation Criteria:**

The experimental setup is well-designed with varying model sizes, training checkpoints, and dataset choices.

**Other Comments Or Suggestions:**

See below questions.

**Other Strengths And Weaknesses:**

See above comments.

**Questions For Authors:**

1. How well do these findings generalize to architectures beyond transformers?
2. What are the computational savings when using this method vs. naive scaling?

**Relation To Broader Scientific Literature:**

This paper builds on scaling law research from Kaplan et al. (2020) and Hoffmann et al. (2022), but extends their findings to diverse model families. Unlike prior studies that focused on fixed model families, this work demonstrates that cross-family extrapolation is feasible, opening new possibilities for scaling predictions across architectures.

**Theoretical Claims:**

The functional form of scaling laws follows prior work (Hoffmann et al., 2022). The paper provides empirical validation for using intermediate checkpoints but lacks a formal theoretical justification. The trade-offs between training fewer large models vs. many small ones are explained well empirically, but a formal learning-theoretic analysis would strengthen the argument.

---

> ### Author Rebuttal · Authors · 2025-04-01
>
> We thank the reviewer for the positive feedback and the accurate points made are encouraging that some deep points about scaling laws indeed pass to readers.
>
> Regarding the two questions:
> 1. How well do these findings generalize to architectures beyond transformers?
> Our experiments focused on two classes of Transformer models (encoder–decoder and decoder-only). Scaling laws for mixtures of experts seem to work relatively well (while not reported in the paper, we do have MoEs included in the scaling law dataset that we will release). We also tested some small Mamba models; they were noisy as they didn’t have as many checkpoints and sizes, but they generally seemed to behave in the same way (in the sense that fitting laws behaved similarly). However, those are only anecdotes and not a robust study.
> 1. What are the computational savings when using this method vs. naive scaling?
> As we provide a guide, it depends on individual time, cost, and hardware constraints on training. For example, if you just train on the beginning of your largest model, a 3x speedup is achievable but requires the same hardware as the large model (see e.g. Fig. 6). If you train on many small models then it depends on how well you need your estimation to be, look for example for the orange lines in the main figure (Fig. 2), they highlight places with approximately the same amount of compute, so you can see for example that often you save more than any amount of training on the large model (so more than 10X), usually, much more. Note that an additional benefit from running smaller models is the hardware, one can use smaller GPUs, less memory and other hardware that is easier to come by, in addition to the FLOPs saved.
>
> We thank the reviewers again for the appreciative review and thoughtful comments. It seems like no major issues preventing the paper from being accepted. We humbly ask if they can raise their score to signify it to the AC.

---

### Official Review · Reviewer_Hw8m · 2025-03-15

**Overall Recommendation:** 3

**Summary:**

The paper is a meta-analysis of scaling law fitting, focusing on a popular power law relating the pretraining loss to the model size and number of training tokens. The paper offers guidelines for how accurate these scaling laws can and should be and how to best estimate them given available resources.

## Update after rebuttal

The authors have addressed my primary concern, and I have raised my score accordingly.

**Claims And Evidence:**

The paper adequately supports its claim with evidence. ~However, I am concerned that a flaw in the analysis (described below under “Experimental Designs Or Analyses”) might invalidate the evidence.~

**Essential References Not Discussed:**

I am missing a more detailed comparison between the dataset of experiments results compiled in this paper and the ones collected for recent meta-analyses such as Ruan et al. (2024) and Maia Polo et al. (2024). Such comparison would be valuable for future practitioners trying to decide which compilation to use in their studies.

**Experimental Designs Or Analyses:**

~I have concerns about the method used to fit the central power law in this paper. In particular, it appears to miss two crucial components used in most prior work:~
1. ~A positivity constraint on $A,B$ and $E$, implemented in prior work by fitting the log of the scaling exponents and coefficients. Alarmingly, Figure 3 shows $E$ fitted to negative values for some model families!~
2. ~A grid search for the initialization to L-BFGS is important for preventing convergence to suboptimal local minima. The authors make no indication of using this technique in their scaling law fit.~

 (Addressed by author rebuttal)

**Methods And Evaluation Criteria:**

The parametric form (1) at the center of the paper has limitations. In addition to potentially being incorrectly or over-parameterized as discussed in Sec. 9 of the paper, eq. (1) is also difficult to fit - As discussed in Besiroglu et al. (2024), Hoffmann et al.’’s fit for (1) had multiple issues, leading to incorrect prediction of compute optimal scaling. The two other approaches studied in Hoffman et al. sidestepped the need to assume (1) holds and instead directly predicted the compute optimal scaling, leading to consistent results validated by subsequent work.

More broadly, pretraining loss is arguably not the most important thing to predict using scaling - predicting optimal design choices such as model size, number of experts, and dataset composition as a function of compute budget is operationally much more meaningful than predicting the resulting loss of the model. Consequently, it would have been more interesting to study how the accuracy of those predictions varies with the specifics of the training procedure.

However, I acknowledge that the parameter form (1) is popular in the literature, and therefore I do not consider focusing on it a critical flaw in the paper.

**Other Comments Or Suggestions:**

The definition of model family in line 74 (right) is not clear. What does it mean for models to “differ only in size?” Even for standard Transformer models, models of different sizes necessarily differ by other hyperparameters such as depth, d_model, attention heads, etc. I think what you meant is that a family is a mapping from (model size, num tokens) to model checkpoints.

Also, I really liked the title of the paper.

**Other Strengths And Weaknesses:**

Strength: The publicly released compilation of scaling experiments results collected as part of this paper could be useful for subsequent work. However, most of the files in the anonymous repository linked to the papers were not available, which preventing me from gauging the usability of the dataset.

**Questions For Authors:**

~My main question is whether the fitting of the scaling law is valid, given the concerns I raised under “Experimental Designs Or Analyses” above.~

**Relation To Broader Scientific Literature:**

The paper revisits the fitting of the power law (1), but unlike prior work it uses a large and diverse body of scaling experiment results, and experiments with degrees of freedom like using intermediate checkpoints from training, and transferring scaling laws between model families, which to my knowledge were not explicitly explored in prior work.

**Theoretical Claims:**

N/A.

---

> ### Author Rebuttal · Authors · 2025-04-01
>
> We thank the reviewer for the deep read, care for details and supportive stance.
>
> The main concern raised by the review had to do with the estimation of scaling law parameters (and why some were reported as negative).  As discussed below, this was (fortunately) a mistake in our presentation rather than the estimation procedure itself.
>
> **Estimation of scaling parameters:**
> Great catch! Indeed, revisiting the graph, the caption does not mention that we have the positivity constraint. Following past work, we fit (and plot) log E, not E, to ensure positivity. We will clarify this in our revised version of the paper.
>
> Similarly, we will clarify that when we fit with L-BFGS (which we found to be less stable as stated in “Fit” Section), we search the parameters using the code from Muennighoff (2023) (which is improved over Hoffman).
>
> We also discuss other mentioned comments.
>
> **Parametric form has limitations:**
> As the reviewer states, “(1) is popular in the literature”; and as we state in the paper, this is our reason to build upon it for this meta-study. Based on preliminary experiments,  expect that many of our findings would be similar in other functional forms.
>
> Note that, while previous work focuses on identifying new functional forms for scaling laws, we here are analyzing how to fit these laws by evaluating generalization on held-out models. Consequently, our results report the extent to which the scaling laws generalized under different settings and allow others to know what to expect.
>
> Besiroglu (2024) notes issues with the fitting reported in the original paper, but those are related to rounding and other issues that are not relevant to our case, especially not with the repetitions across settings, that do not share the same issues.
>
> **Pretraining loss vs. downstream task:**
> You are correct that various other kinds of quantities can be predicted with scaling lows. We believe this would be an excellent topic for a follow-up paper!
>
> **Comparisons to observational scaling law papers:**
>
> We are happy to add more discussion of the relationship of this work to those! We value their contribution to the field (as can be seen by citing them in various contexts). We wish to emphasize that our papers have different goals: Ruan focuses on predicting scaling across datasets  and Polo proposes a new kind of scaling laws that fits across multiple models together; both given fixed data for estimating a scaling law. Here we’re focused specifically on identifying *what data to collect*---a question that could also be asked about observational laws in future work. Our approach does not provide a new scaling law, but discusses fitting scaling laws, when creating new models, efficient use of the information available etc. Possibly, information from previous models can aid there as well (but it is unclear, as the distinctions in A/B tests might be smaller than the ones between full models that they focus on).
>
> Model size:
> Thank you, we will clarify based on your suggestions.
>
> Title:
> Thanks, that’s reassuring!

---

> > ### Comment · Reviewer_Hw8m · 2025-04-05
> >
> > Thank for addressing my concern regarding parameter fitting - if the issue of negative E was a missing log factor then I think the paper passes the bar for publication and will update my evaluation accordingly.
> >
> > Regarding comparison to Ruan and Polo - I was asking for a _dataset_ comparison: how does you compilation of model family evaluation differs from theirs?

---

> > > ### Author Response · Authors · 2025-04-07
> > >
> > > Regarding data, the main difference is that their data uses benchmark scores (so not the data scaling laws are usually having, but the data evaluation is using). This has a lot of benefits (e.g., high data availability for other models), but also disadvantages (e.g. that scaling law and learning trajectories research cannot use it). Our data includes either losses or downstream (or both) but it is collected along the training run and not just at the end. Note that the end also has interesting but different characteristics such as instruction tuning, changing data at the last next token prediction batches, changes in hyperparameters and such behaviors that we try to isolate and their data tries to see as a whole. Both are valuable, but for different scientific questions. When available, we also match those results with the actual checkpoints released to allow further research of this kind.

---

### Decision · Program_Chairs · 2025-05-01

**Decision:**

Accept (poster)

**Comment:**

The work collects a dataset containing upstream losses and downstream evaluations for 485 models to facilitate research in scaling law estimation and validation. Based on these, they come up with a recommended set of best practices. The focus here is on extrapolating performance from small models to large ones. Among other findings, the authors argue that with a single model from a target family and a scaling law from a different model family, it is sometimes possible to accurately estimate target model performance reasonably well. Also, because one can reduce variance by training many models, training many small models and averaging their results can give more accurate scaling laws than including larger models.

Overall, reviewers found the paper to be a valuable contribution to the community. However, I urge the authors to revise the paper in light of the reviewers' feedback.

For example, while the authors claim that their results are robust to the training hyperparameters, some reviewers raised concerns about this. I think this assumption is tricky. For example, switching to a cosine learning rate schedule often results in a loss curve that violates the power law assumption (with a rapid improvement near the middle of training). There are of course other popular lr schedules that violate the power-law assumption even more, such as decaying the lr by a factor of 10 at various points during training (see for example https://arxiv.org/pdf/1912.11370). The authors should be cautious in claiming generality of findings.

Second, since one primary contribution of this work is releasing a benchmark dataset, the authors should mention similar past efforts. See for example the benchmarks released in [1] and [2].  It appears that in the current version of the paper, there is no mention of the fact that other benchmarks exist.

[1] Besiroglu, T., Erdil, E., Barnett, M., & You, J. (2024). Chinchilla scaling: A replication attempt. arXiv preprint arXiv:2404.10102.

[2] Alabdulmohsin, I. M., Neyshabur, B., & Zhai, X. (2022). Revisiting neural scaling laws in language and vision. Advances in Neural Information Processing Systems, 35, 22300-22312.